# The *Chloranthus sessilifolius* genome provides insight into early diversification of angiosperms

Jianxiang Ma[1,3], Pengchuan Sun [2,3], Dandan Wang[1,3], Zhenyue Wang[1], Jiao Yang[1], Ying Li[1], Wenjie Mu[1], Renping Xu[1], Ying Wu[1], Congcong Dong[1], Nawal Shrestha[1], Jianquan Liu[1,2] & Yongzhi Yang [1✉]

Most extant angiosperms belong to Mesangiospermae, which comprises eudicots, monocots, magnoliids, Chloranthales and Ceratophyllales. However, phylogenetic relationships between these five lineages remain unclear. Here, we report the high-quality genome of a member of the Chloranthales lineage (*Chloranthus sessilifolius*). We detect only one whole genome duplication within this species and find that polyploidization events in different Mesangiospermae lineage are mutually independent. We also find that the members of all floral development-related gene lineages are present in *C. sessilifolius* despite its extremely simplified flower. The *AP1* and *PI* genes, however, show a weak floral tissue-specialized expression. Our phylogenomic analyses suggest that Chloranthales and magnoliids are sister groups, and both are together sister to the clade comprising Ceratophyllales and eudicots, while the monocot lineage is sister to all other Mesangiospermae. Our findings suggest that in addition to hybridization, incomplete lineage sorting may largely account for phylogenetic inconsistencies between the observed gene trees.

[1] State Key Laboratory of Grassland Agro-Ecosystems, Institute of Innovation Ecology and School of Life Sciences, Lanzhou University, Lanzhou, China. [2] Key Laboratory of Bio-Resource and Eco-Environment of Ministry of Education & State Key Laboratory of Hydraulics & Mountain River Engineering, College of Life Sciences, Sichuan University, Chengdu, China. [3] These authors contributed equally: Jianxiang Ma, Pengchuan Sun, Dandan Wang. ✉email: yangyongzhi2008@gmail.com

Angiosperm diversification has produced the most spectacular species biodiversity in terrestrial ecosystems[1,2], providing basic necessities, including food, clothing fibers, timber, medicine and fuelwood for humans, and major ecological services, including photosynthesis and carbon sequestration[3,4]. Except for Amborellales, Nymphaeales, and Austrobaileyales (collectively known as ANA-grade), which only includes ~175 species[5], the vast majority (~99.95%) of extant angiosperms belong to Mesangiospermae, and can be classified into five major lineages: eudicots, monocots, magnoliids, Chloranthales and Ceratophyllales[6]. Eudicots and monocots are the two largest and the most diverse of these lineages, respectively including around 75 and 22% of all species[7]. Magnoliids comprise 10,000 species in four orders: Canellales, Laurales, Magnoliales, and Piperales[6,8]. In contrast, Chloranthales and Ceratophyllales are small lineages, with only 77 and 4 extant species, respectively[2]. Both lineages have unusual morphological characters and both are important for understanding phylogenetic relationships among the major angiosperm lineages[2,7,8].

Based on fossil records, angiosperms were suggested to originate approximately 140 million years ago (Mya) followed by a rapid diversification[9–13]. The sudden appearance of the diverse angiosperm fossils within a very short geological period, also known as the "Darwin's abominable mystery"[14,15], makes it difficult to disentangle the phylogenetic relationship among these early-diverged lineages. The existing molecular phylogenies resolve Amborellales, Nymphaeales, and Austrobaileyales as successive sisters to Mesangiospermae[16–18]. However, phylogenetic relationships among the five major Mesangiospermae lineages remain uncertain and multiple alternative topologies have been proposed[19–25]. For example, based on plastid genes, a trifurcation topology has been proposed for Chloranthales, magnoliids and (monocots + [eudicots + Ceratophyllales])[24], but a sister relationship between Chloranthales and magnoliids by others[23]. A recent phylogeny based on 2881 plastid genomes placed Chloranthales as the earliest diverged lineage of Mesangiospermae, with magnoliids and monocots successively sister to Ceratophyllales and eudicots[19]. However, a transcriptome-based phylogeny of 60 angiosperms taxa[26] placed Chloranthales as a sister group to eudicots-Ceratophyllales. The OneKP Project[21], which was also based on transcriptome data, however, recovered the sister relationship between Chloranthales and magnoliids. Some genomic studies of magnoliids supported a sister relationship between magnoliids and eudicots[27–30], while others supported magnoliids sister to the other Mesangiospermae lineages[31–33]. The recently reported non-duplicated magnoliid genome of Aristolochia fimbriata supported a sister relationship between magnoliids and monocots based on shared fusion events[34]. On the other hand, phylogenomic analyses based on the Ceratophyllum genome supported a sister relationship between Ceratophyllales and eudicots, which together were recovered as sister to magnoliids and further suggested that incomplete lineage sorting (ILS) likely accounts for some phylogenetic discordances[17]. However, the lack of Chloranthales genome has greatly limited our understanding of phylogenetic relationships and early diversification of these angiosperm lineages.

Chloranthus sessilifolius ($2n = 2\times = 30$, Chloranthaceae; Fig. 1a)[35] is a wild diploid aromatic herb, which produces very simple flowers with only three androecial lobes, three stamens and one pistil[36,37]. All Chloranthus plants have rich volatile compounds that mainly contain sesquiterpenoids and diterpenoids[37]. In addition, only scalariform perforation plates, rather than well-developed vessels, are found in Chloranthus[35]. Many chloranthoid pollen fossils (e.g., Hedyosmum, Asteropollis, etc.) were recovered dating back to the early Cretaceous[13], suggesting a widespread distribution of Chloranthales since the early

Cretaceous. Genome sequences may provide us important cues to understand the special traits of Chloranthus and resolve the evolutionary relationship among the Mesangiospermae lineages.

Here, we report the high-quality chromosome-level reference genome of C. sessilifolius using Illumina short reads, Oxford Nanopore Technologies (ONT) long reads, and Hi-C sequencing. The availability of genomes for the representatives of the other four main Mesangiospermae lineages made it possible to carry out comprehensive evolutionary analyses using whole genome data. We detect one whole genome duplication within C. sessilifolius and find that the polyploidization events in each Mesangiospermae lineage are mutually independent. Our analyses reveal a sister relationship between Chloranthales and magnoliids, and the highly discordant gene trees between five Mesangiospermae lineages. We deduce that both hybridization and incomplete lineage sorting may have together contributed to such phylogenetic incongruities.

## Results and discussion

**Genome assembly and annotation of C. sessilifolius.** We generated 100 Gb of Illumina short reads from genomic DNAs obtained from young leaves of C. sessilifolius, recovering the species' genome size of 2232.26 Mb (Supplementary Fig. 1). We then generated 207.2 Gb (95.43 × depth) of high-quality long reads (N50 length, 36.4 kb) using a Nanopore platform (Supplementary Table 1). The C. sessilifolius genome was initially de novo assembled and then polished by four rounds of Illumina short reads. The resulting genome spanned 2168.73 Mb with a contig N50 of 53.74 Mb (Supplementary Table 2), constituting 97.15% of the estimated genome size. These contigs were further assigned to 15 pseudo-chromosomes (with sizes ranging from 95.71 to 199.17 Mb) by Hi-C analysis, and ~99.43% of the assembled sequences could be properly anchored (Supplementary Table 3 and Supplementary Fig. 2). We further assessed the quality of the C. sessilifolius genome and we found that more than 99.93% of Illumina short reads could be mapped to the assembly (Supplementary Table 2). The GC content and sequencing coverage both had a Poisson distribution (Supplementary Fig. 3), more than half the length of 97.65–99.21% of de novo assembled transcripts could be mapped to one contig (Supplementary Table 4), and ~92.4% of 1375 BUSCO genes (Embryophyta odb10) could be completely predicted in our assembly (Supplementary Fig. 4a). We conclude that our C. sessilifolius genome has high degrees of both accuracy and completeness (Fig. 1b).

A total of 34,065 protein-coding genes were predicted in C. sessilifolius with an average CDS length, exon length and exon number of 1195.18, 202.34, and 5.91, respectively, similar to reported parameters for other angiosperm species (Supplementary Table 5 and Supplementary Fig. 5). Nearly 97.65% of the genes could be assigned to entries in five functional databases by Blast searches (Supplementary Table 6). BUSCO analysis further revealed that 1255 complete genes (91.3%) were present in our predicted gene set, indicating that most of the gene models were complete (Supplementary Fig. 4b). The non-coding RNAs were further identified, which included 889 tRNAs, 767 rRNAs, 296 miRNAs, and 7827 snRNAs in the assembled genome (Supplementary Table 7).

We also detected a total of 1.41 Gb (64.94%) transposable elements (TEs) in the C. sessilifolius genome (Supplementary Table 8). Long terminal repeats (LTRs) retrotransposons were predominant, accounting for 54.81% of the whole genome. However, the distribution of TEs was asymmetric along the genome with significantly high accumulation within intergenic than genic regions (Supplementary Fig. 6 and Supplementary Table 9) possibly due to their potentially detrimental effects on

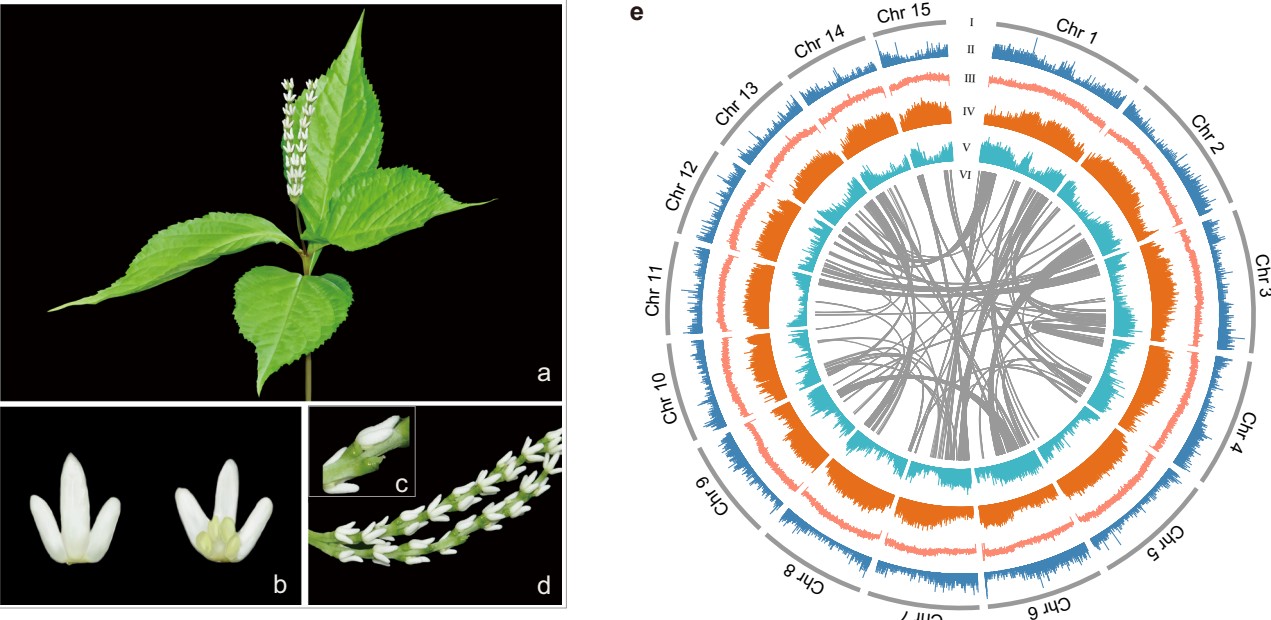

**Fig. 1 *C. sessilifolius* morphology and genome features. a** Habit of *C. sessilifolius*. **b** Stamen. **c** Pistil. **d** Inflorescence. **e** Overview of *C. sessilifolius* genome. Different tracks (moving inward) denote (I) chromosomes; (II) gene density in 100 kb sliding windows (minimum–maximum, 0–30); (III) GC content in 500 bp sliding windows (minimum–maximum, 0.2–0.8); (IV) *Gypsy* density in 10 kb sliding windows (minimum–maximum, 0–1.0); (V) *Copia* density in 10 kb sliding windows (minimum–maximum, 0–1.0); (VI) identified syntenic blocks. Source data are provided as a Source Data file.

gene expression[38]. We also detected a tendency for TEs to accumulate frequently in introns within the genic regions, which therefore resulted in long introns in *C. sessilifolius* (Supplementary Fig. 6 and Supplementary Tables 5 and 10). We also observed relatively longer genes (> 20 kb, 8187 genes) in *C. sessilifolius* than in other species (Supplementary Fig. 7a), and this is a common feature of the large genomes[28,39]. Moreover, TE contents of the genes' introns were positively correlated with the introns' length in *C. sessilifolius* ($R^2 = 0.18$, $p < 0.001$, Supplementary Fig. 7c). Exploration of the historical dynamics of full-length Gypsy and Copia retrotransposons in this species indicated that both apparently proliferated less than 15 million years ago (Mya) but was earlier for Gypsy than Copia (Supplementary Fig. 7d).

**Polyploidization histories of *C. sessilifolius* and other representatives**. We used multiple methods to explore polyploidization histories of *C. sessilifolius* and other representative species from the major angiosperm lineages. The distributions of synonymous substitutions per synonymous site (*Ks*) of homolog pairs from intragenomic and intergenomic syntenic blocks were estimated. We detected obvious signs of one polyploidization event in *C. sessilifolius* (*Ks* peak of ~1.07, Fig. 2a and Supplementary Fig. 8). This event occurred more recently than the divergence between *C. sessilifolius* and other species, but was similar to the divergence time between *C. sessilifolius* and *Liriodendron chinense*, which may suggest a close relationship between *C. sessilifolius* and magnoliids (Supplementary Fig. 9). Consistent with previous studies[16,17] we also detected one or multiple polyploidization events in other species (Fig. 2a and Supplementary Fig. 8). To better elucidate the polyploidization history of *C. sessilifolius*, we further performed the intragenomic and intergenomic syntenic analyses. Within the *C. sessilifolius* genome, one-to-one syntenic blocks are predominant (Fig. 2c). However, only a few large collinearity segments were detected, for example, one between chr 2 and the tail of chr 6, and another between chr 9 and the head of chr 6 (Fig. 2c). Such patterns indicate that an ancient whole genome duplication (WGD) might have occurred in *C.*

*sessilifolius*, but followed by chromosomal breaks, fusions and gene losses[40]. Further intergenomic syntenic analyses between *C. sessilifolius* and *Amborella trichopoda*, *L. chinense*, *Vitis vinifera*, obtained syntenic depth ratios of 2:1, 2:2, and 2:3, respectively (Fig. 2d and Supplementary Fig. 10), corroborating all the analysis and suggesting that only one WGD occurred in the evolutionary history of *C. sessilifolius*.

We performed further phylogenetic analyses to determine if the WGD occurring within *C. sessilifolius* was independent or shared by other species. Collinear genes between *C. sessilifolius* and the other nine species were extracted and used to build the gene trees. Our analyses included two Nymphaeales species (*Euryale ferox* and *Nymphaea colorata*), two magnoliids (*Cinnamomum kanehirae* and *Liriodendron chinense*), one monocot (*Elaeis guineensis*), three eudicots (*Aquilegia coerulea*, *Prunus persica*, and *Vitis vinifera*) and one Ceratophyllales (*Ceratophyllum demersum*). We found that most collinear gene trees (62–97%) well supported the independent WGD event for *C. sessilifolius* (Fig. 2b). These results also supported that all detected polyploidization events in each Mesangiospermae lineage were mutually independent (Fig. 2b). The collinear gene tree analyses (Fig. 2b) also suggested that *Cinnamomum* and *Liriodendron* shared one WGD event[17,28,29,34,41] and this was also confirmed by the highly conserved gene arrangements detected between Laurales and Magnoliales (Fig. 2e and Supplementary Fig. 11).

**Phylogenetic relationships of Chloranthales and other angiosperms**. The high-quality of the *C. sessilifolius* genome allowed examination of the phylogenomic relationships of the five Mesangiospermae lineages. First, a set of 1689 single-copy orthologous genes were identified with SonicParanoid[42] using genomes of 14 plants, which included one gymnosperm (*Ginkgo biloba*) as the outgroup, three species from the ANA-grade (*A. trichopoda*, *Euryale ferox*, and *N. colorata*), two magnoliids (*L. chinense* and *Cinnamomum kanehirae*), three monocots (*Oryza sativa*, *El. guineensis*, and *Apostasia shenzhenica*), three eudicots (*Aquilegia coerulea*, *Prunus persica*, and *V. vinifera*), *C.*

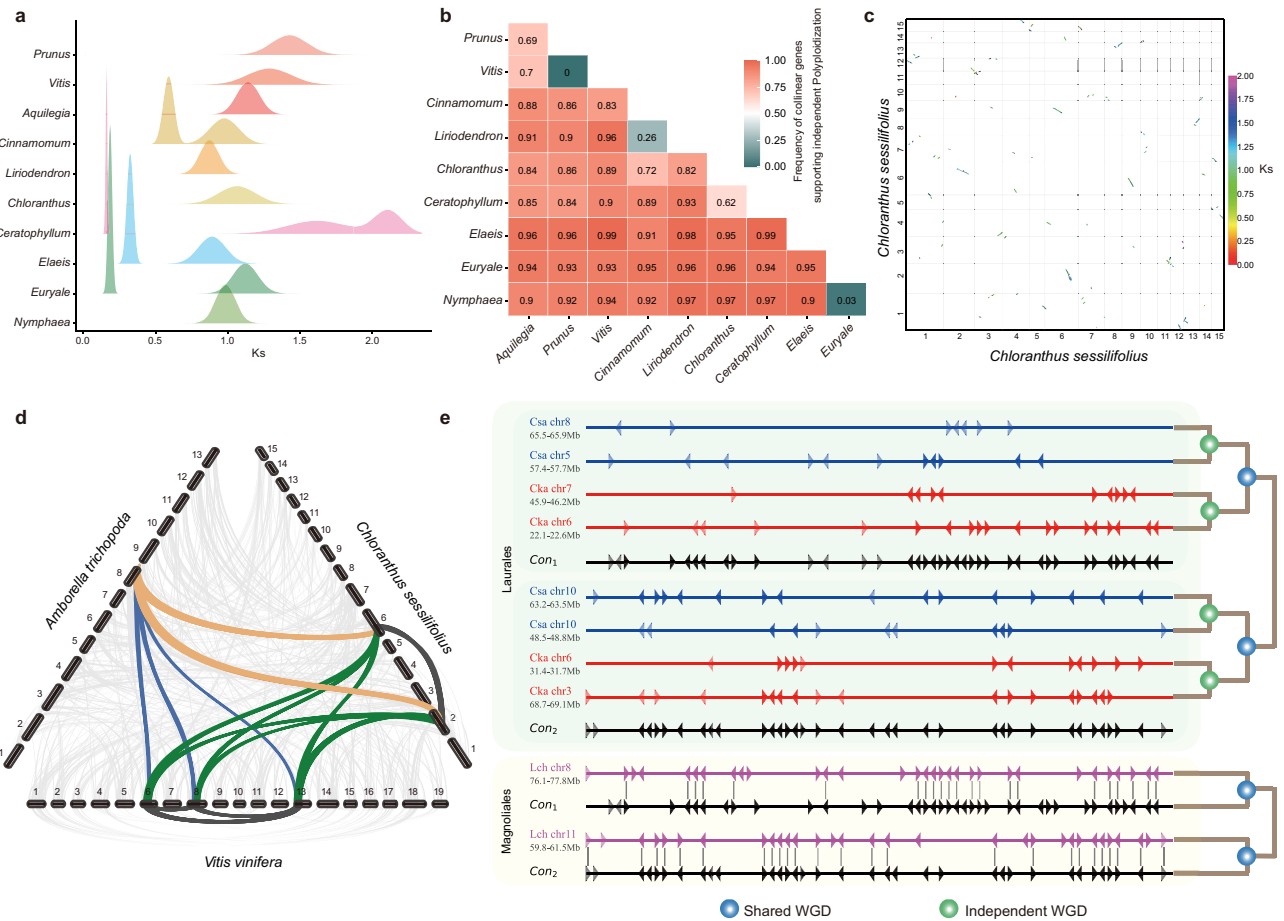

**Fig. 2 Comparative genomics analyses. a** The Ks distributions of intragenomic synteny blocks. **b** The proportion of collinear gene trees that support the independent polyploidization between each species-pair. **c** Synteny blocks of the *C. sessilifolius* genome. **d** Synteny patterns between genomic regions from *C. sessilifolius, A. trichopoda,* and *V. vinifera.* **e** Local alignment of Magnoliales (*Liriodendron*-Lch) and Laurales (*Cinnamomum*-Cka and *Chimonanthus*-Csa). The graph shows details of a short segment of alignment, and the names behind the species are the chromosome numbers. The location and direction of genes are shown by a triangle. Reconstructed putative ancestral chromosome segments of Laurales, named as Con1 and Con2, are displayed accordingly. Homologous genes between neighboring chromosomal regions are linked with lines. Source data are provided as a Source Data file.

*sessilifolius* (Chloranthales) and *Ce. demersum* (Ceratophyllales) (Supplementary Table 11). A highly supported species tree was obtained through maximum-likelihood (ML) analysis of the concatenated nucleotide sequences (Fig. 3a and Supplementary Fig. 12). This phylogenetic tree supported the hypothesis that Chloranthales is sister to magnoliids, rather than other Mesangiospermae lineages. The magnoliids + Chloranthales clade was sister to the eudicots + Ceratophyllales clade, while the monocot lineage was sister to other Mesangiospermae lineages. We further applied coalescent-based phylogenetic analysis in ASTRAL using each gene tree, and yielded the same topology with high posterior probabilities (Fig. 3a and Supplementary Fig. 12). Our inferred topology is consistent with the recent phylogenomic analyses[21,22,43], which used a relatively smaller single-copy gene set than our study. On the other hand, our topology also differs from a few studies, which either recovered magnoliids as sister to other Mesangiospermae lineages (based on phylogenomic analyses)[32–34], or revealed the sister relationship between monocots and magnoliids (based on chromosome fusion events)[34]. So, in order to improve the accuracy of our phylogeny, we firstly used TreeShrink[44] to remove sequences that may lead to unrealistically long branch lengths and the results were highly consistent (Supplementary Fig. 13). To avoid the influence of methodological orthology inference and outgroups, OrthoMCL

was further employed to extract single-copy orthologous genes (designated OSCGs) and low-copy genes (LCGs) with alternative outgroup (*Picea abies*). Based on the 866 OSCGs extracted, both concatenation and coalescent phylogenetic analyses produced results consistent with the former tree (Supplementary Fig. 12). The large dataset of 2097 LCGs was further used with two methods and yielded consistent topologies (Fig. 3a and Supplementary Fig. 14). In addition, we used collinear genes to construct species trees. We only selected species that have chromosome-level assemblies and that show clear polyploidization history of the mentioned former 14 species. As a result, we selected 11 species and excluded three species: *Ginkgo biloba, Apostasia shenzhenica,* and *Oryza sativa.* This method can eliminate errors in orthology inference[45,46], especially during gene family clustering. Using *Amborella* as a reference, we assigned the syntenic blocks into different copies in each species according to its polyploidization history (Supplementary Figs. 15–17). Only genes that have a collinear relationship with *Amborella* and have at least eight species were retained. A total of 4120 collinear genes were retrieved to infer the species tree based on the coalescent method. This synteny-based species tree showed the same topology as the former tree, and also clearly reflected the polyploidization history of each species consistent with the previous polyploidization analyses (Figs. 2 and 3b and Supplementary Fig 18).

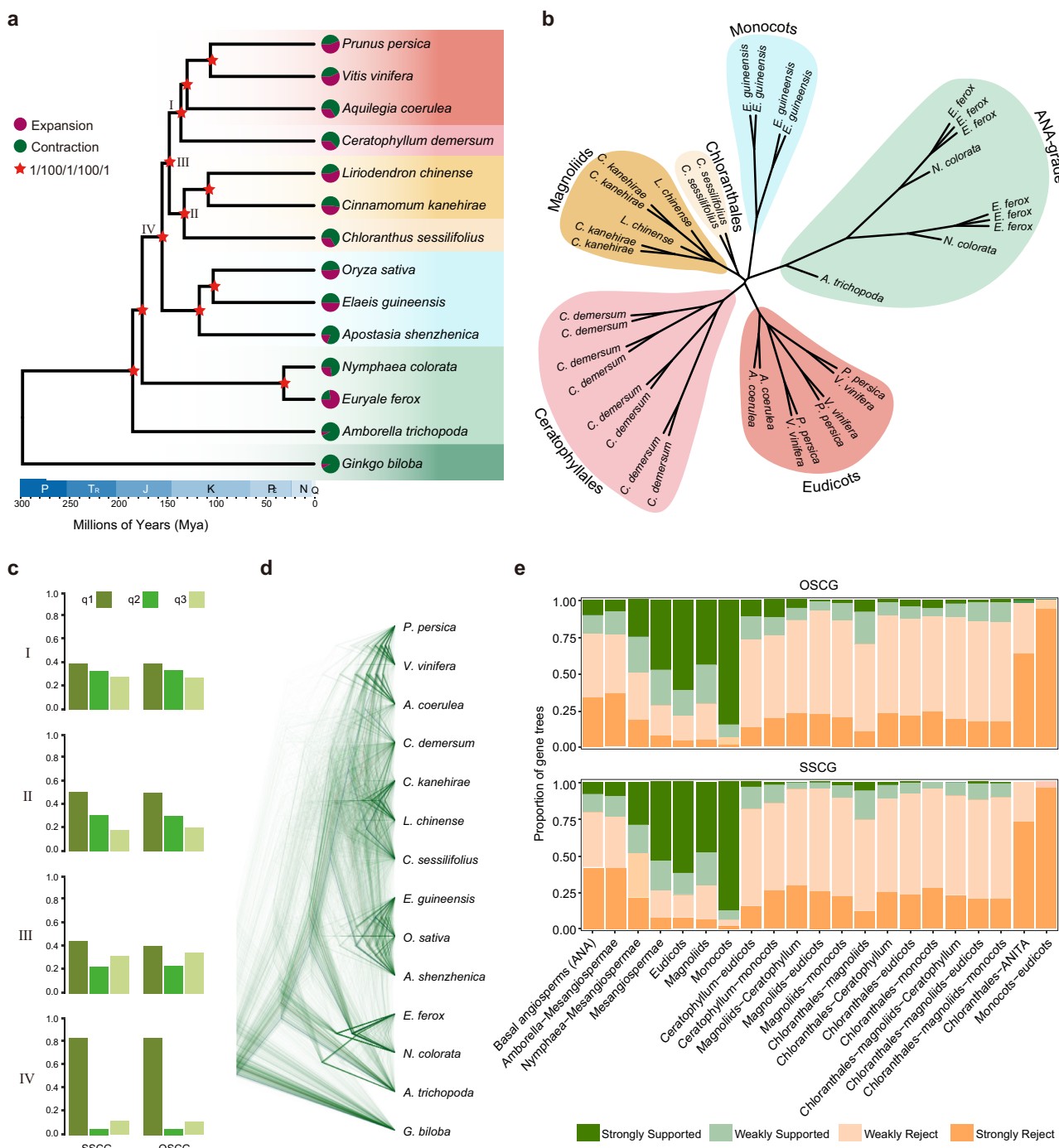

**Fig. 3 Phylogenomic analysis of major groups of angiosperms. a** Phylogenetic tree of 14 species based on nucleotide sequences of five datasets. Estimated divergence times and the time scale are shown at the bottom. Bootstrap support (BS) values and posterior probabilities (PP) are indicated with a red asterisk for each internal branch (from left to right: multi-species coalescent-based (PP), concatenated-based (BS), multi-species coalescent-based (PP), concatenated-based (BS), and multi-species coalescent-based (PP), using SSCG, SSCG, OSCG, OSCG, and LCG datasets, respectively). The pie graph at the end of the terminal branches represents the proportion of gene families that underwent expansion (red) or contraction (green) when comparing with their most recent common ancestor. **b** Synteny-based phylogeny tree of the selected 11 species (see the detail in Supplementary Fig. 18). **c** The proportions of gene trees with different topologies. The focal internal branches are marked with I, II, III, and IV. q1, q2 and q3 indicate the quartet support for the three alternative topologies. **d** Superimposed ultrametric gene trees based on the SSCG dataset. **e** Gene tree compatibility. The proportion of gene trees for which focal splits are highly (or weakly) supported (or rejected) are shown in respective colors. Weakly rejected splits are those that are not in the tree but are compatible if low (<75%) support branches are contracted. Source data are provided as a Source Data file.

**Phylogenetic discordance and possible causes.** We identified the obvious discordance between the nuclear and plastome phylogenies. In contrast to the nuclear phylogeny, the plastome phylogeny placed the clade comprising Chloranthales and magnoliids sister to other Mesangiospermae lineages (Supplementary Fig. 19). We also detected widely conflicting topologies between nuclear gene trees. Densitree clearly revealed the high discordant tree topologies between four of the five main lineages: eudicots, magnoliids, Chloranthales, and Ceratophyllales based on different nuclear genes (Fig. 3d). Most gene trees supported a sister relationship between monocots and all other Mesangiospermae lineages (Fig. 3c, node IV, topology q1). However, different tree topologies were also found, including sister relationships between Ceratophyllales and eudicots, between magnoliids and Chloranthales and between (Ceratophyllales + eudicots) and (Chloranthales + magnoliids) (Fig. 3c, node I, II, III). These discordances summarized by DiscoVista revealed that most gene trees strongly rejected the sister relationship between monocots or eudicots although weakly refuting the sister relationship between each two of the other Mesangiospermae lineages (Fig. 3e). We also discovered that most gene trees strongly refuted the sister relationship between Chloranthales and ANA-grade (Fig. 3e).

Many factors could cause the incongruent tree topologies among nuclear genes or between nuclear and plastome genes[17,27,33]. One of such factors could be the sparse taxon sampling. In order to examine this possibility, we added additional 28 published genomes making a total of 30 angiosperm orders included in our analyses (Supplementary Table 11). A total of 1846 "mostly" single-copy orthologous genes were extracted. The average number of genes per taxon was 1735 (Supplementary Fig. 20). Coalescent analyses using this dataset also recovered consistent phylogenetic relationships among the five major Mesangiospermae lineages similar to the one found in previous analyses (Fig. 4b and Supplementary Fig. 21). Multiple independent polyploidization events (Fig. 2), hybridization, and especially allopolyploidization might have led to such phylogenetic discordances. We, therefore, examined this possibility using PhyloNetworks[47]. We detected three likely hybrid events between monocots and *Nymphaea*, between Chloranthales and the ancestor of eudicots and Ceratophyllales, and between monocots and magnoliids (Supplementary Fig. 22). These possible hybridizations may partly explain the topological discordance between nuclear or plastome phylogenies found herein (Fig. 3 and Supplementary Fig 19) and reported before[16,32,33].

The short divergence intervals between five Mesangiospermae lineages (within 23 Mya, between 158 and 135 Mya based on our estimations, Supplementary Fig. 23) suggests that they diversified within a very short time. Therefore, except for hybridization, incomplete lineage sorting (ILS) might have also occurred during early diversification. We estimated the theta parameter, which reflects the level of ILS[48], for each internal branch by dividing the mutation units inferred by IQ-TREE by coalescent units inferred by ASTRAL. We found that the theta values ranged from 0.027 to 0.224, and the ancestor branch of Ceratophyllales and eudicots showed the highest level of ILS, while the ancestor branch of Chloranthales and magnoliids showed a low level of ILS (0.043) (Fig. 4a). This difference was also detected in the analyses with the increased taxon dataset (Fig. 4b). We further simulated 20,000 gene trees with the ILS conditions by Phybase[49] and DendroPy[50] under the multispecies coalescent model. There was a high correlation between the two simulated datasets (Fig. 4f). The considerable agreement between empirical gene trees and simulated ones with ILS was also detected ($R^2 > 0.98$; Fig. 4d, e). We also simulated the gene trees without ILS (by setting the theta value of 0.001 in Phybase), and these trees showed a relative low agreement with our empirical trees ($R^2 = 0.779$, Fig. 4c),

suggesting that ILS could not be excluded to account for topological discordance of gene trees between five Mesangiospermae lineages. We also examined the possibility of ILS in the internal branches based on the chi-square test of the frequency of two minor topologies between empirical data and simulated data with ILS[48]. We found that the phylogenetic discordance of the ancestor branch of Ceratophyllales and eudicots could be totally explained by the ILS effect, while for the other internal branches between five Mesangiospermae lineages, hybridization and other factors could not be excluded to account for the observed inconsistencies in the trees (Supplementary Fig. 24).

**Floral-development related genes in *C. sessilifolius*.** *C. sessilifolius* bears an extremely simple bisexual flower, with only androecial lobes united at the base[37]. We examined the presence and expression of orthologs of the floral development related genes (FDRGs) included in the Flowering Interactive Database (FLOR-ID)[51] in *C. sessilifolius* and other angiosperm lineages. We detected a comparable number of FDRGs in *C. sessilifolius* compared to other sampled species, and eudicots usually contained more FDRGs (Fig. 5a). We also found that the number of these FDRGs agreed with frequencies of putative WGD events (Supplementary Fig. 25). We subsequently focused mainly on MADS-box transcription factors, which are important regulators of flower development. In total, 58 putative MADS-box genes were identified within the *C. sessilifolius* genome with phylogenetic distributions across all major lineages identified for eudicots (Supplementary Fig. 26 and Supplementary Table 12). Among them, 36 genes belonging to type II were further clustered into 21 lineages, and these lineages were highly consistent with those of type II MADS-box genes in *Amborella*[18]. This indicates that all gene lineages and sub-lineages of the MADS-box evolved and had formed in the ancestor of angiosperms. Similar to *Amborella*, the two previously assumed monocot-specific *OsMADS32*[52] and eudicot-specific *TM8*[53,54] gene lineages have orthologs in *C. sessilifolius*. However, the magnoliids only retains the *TM8* lineage[18,55]. Therefore, the loss of these two FDRG lineages seems to be independent and random across different angiosperm lineages[18,55]. It is interesting that the *FLC* gene lineage appeared only in eudicots, but not even in Ceratophyllales, which is sister to eudicots. Homologs of all floral organ identity genes are found in *C. sessilifolius*, including six *AP1*s (class A), two *AP3*s and one *PI* (class B), three *AG*s (class C), one *SEP1* and one *SEP3* (class E). Classes A, B, C, and E have functions in the development of sepals and petals, petals and stamens, stamens and pistils, and interactions with ABC-function proteins[56]. To gain more insight into the functions of the MADS-box homologs in *C. sessilifolius*, we performed RNA-seq for different tissues (Supplementary Fig. 27) and the expression levels of these genes were determined in the three floral organs (androecial lobes, anther and pistil) and leaves. We found that the two E-class genes (*SEP1* and *SEP3*) showed high expressions in flower organs. In addition, the *SEP1* also showed expression in leaf, xylem and phloem. Among the A-class genes, three of the six *AP1* genes have a high transcriptional activity, which may reflect a functional redundancy (Fig. 5b). The activated *AP1* genes are expressed most strongly and may contribute to the development of the perianth-like lobes (Fig. 5b). In addition to C class genes, *AP1*, *PI*, *SOC1*, *AGL6*, and *AGL32* genes are also strongly expressed in anthers and pistils (Fig. 5b). Within the B-class genes, the *AP3* gene was weakly expressed in all the examined tissues, and the others showed a high expression in flower organs, especially the anthers. The *PI* gene was broadly expressed in all flower organs, leaf, phloem and xylem. We also found the expression of many ABC genes (FPKM > 0) in vegetative organs (Supplementary Table 13).

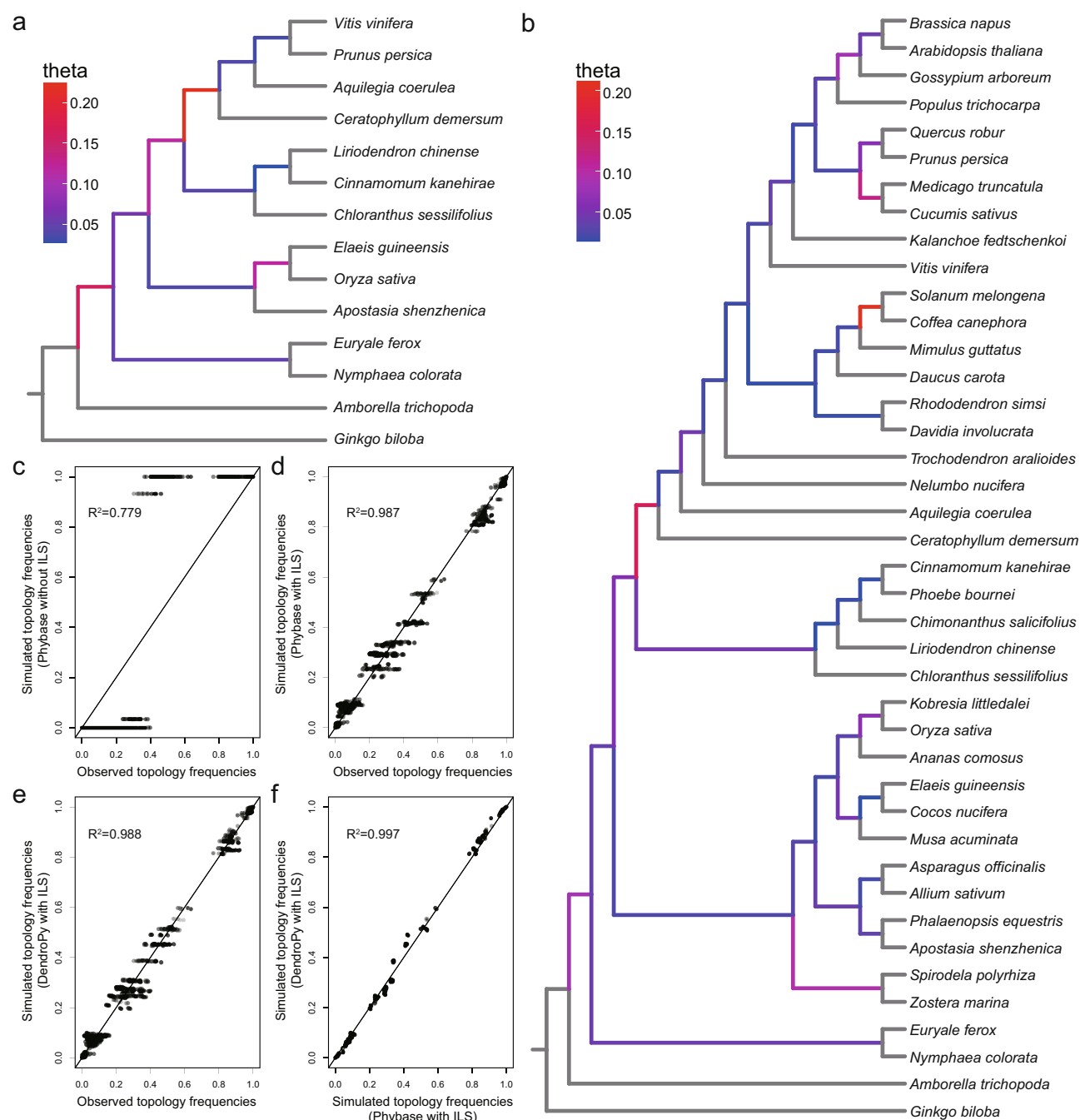

**Fig. 4 ILS analyses. a** Estimated theta value for each internal branch of the 14 species. **b** Estimated theta value for each internal branch of the 41 species. Warmer colors indicate higher theta and thus higher ILS level. Terminal branches are colored gray due to lack of data to infer theta. **c**–**f** represent the correlation analyses of topology frequency of each quartet species combination in the different dataset: the simulation without ILS by Phybase and empirical observation (**c**), the simulation with ILS by Phybase and empirical observation (**d**), the simulation with ILS by DendroPy and empirical observation (**e**), the simulation with ILS by Phybase and DendroPy (**f**). The "lm()" function in R was used to perform all the correlation analyses, and the p values were both less than 0.01. Source data are provided as a Source Data file.

Consistent with our findings, the OneKP also identified the ABC genes from leaf and/or root transcriptomes[21]. From floral tissues, the weak tissue-specific expressions of ABC genes (only *PI* and *AP1* herein) were also reported in previous studies on Nymphaeales[16,57,58] and *Persea americana*[57,58].

**Terpenoid and secondary cell wall biosynthesis genes in *C. sessilifolius*.** *Chloranthus* plants have rich volatile compounds mainly comprising sesquiterpenoids and diterpenoids[59]. To understand the genetic bases of terpenoid biosynthesis in *C.*

*sessilifolius*, we identified a total of 2756 and 5549 chemicals-related gene families that were expanded and contracted in *C. sessilifolius*, respectively (Fig. 3a). Based on the functional enrichment analyses, these expanded gene families were mainly related to terpenoid biosynthesis and metabolic processes, such as "isoprenoid biosynthetic process", "terpenoid metabolic process" and "terpenoid biosynthetic process" (Supplementary Fig. 28 and Supplementary Table 14). In agreement with the result of GO enrichment, the KEGG pathways of terpenoid biosynthesis, flavonoid biosynthesis and phenylpropanoid biosynthesis were also

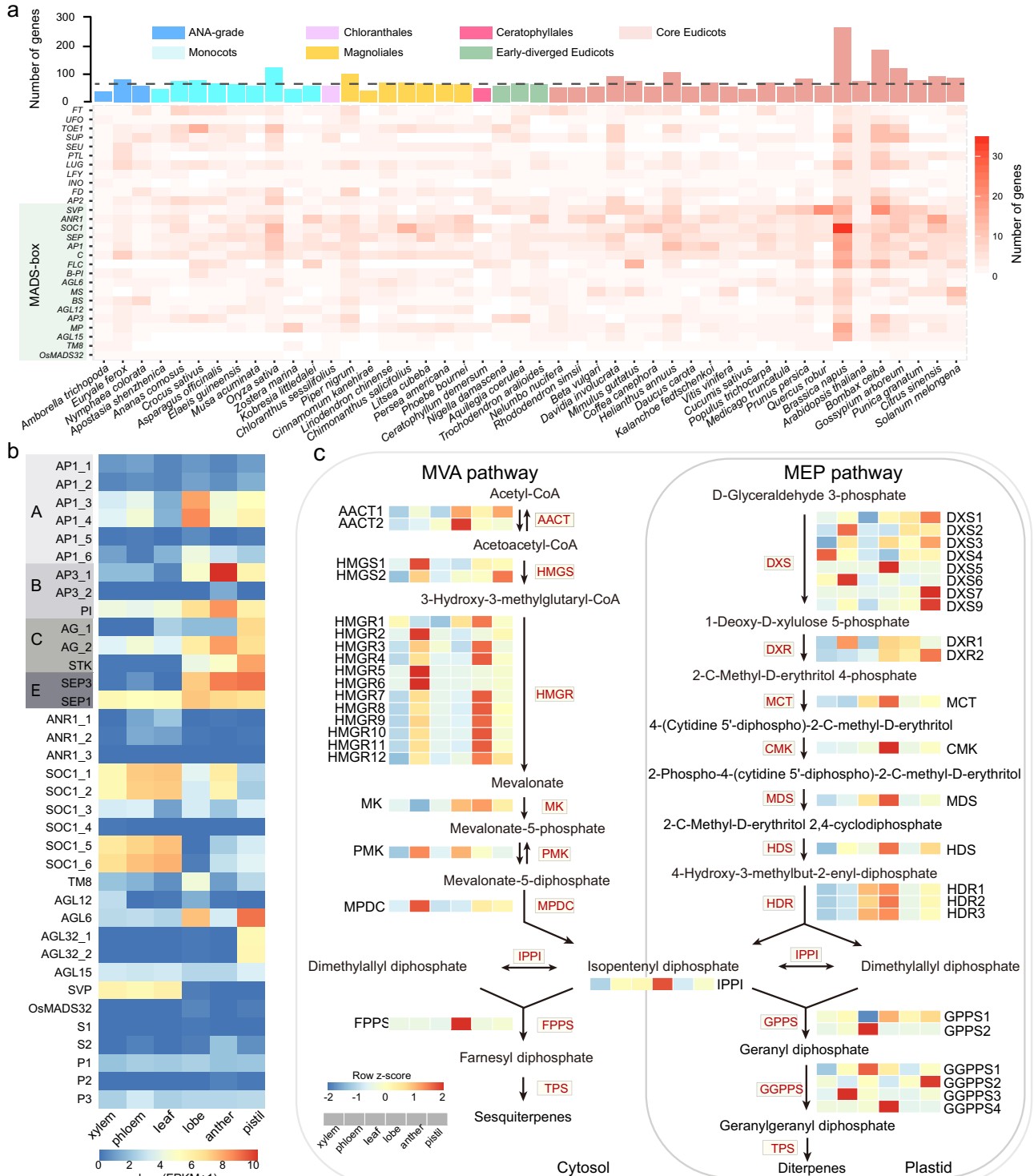

**Fig. 5 Evolutionary analysis of flower development and terpenoid biosynthesis related genes in *C. sessilifolius*. a** Statistics of the numbers of FDRGs (floral development related genes) identified in 46 species. The color in each cell of the heatmap represents the number of the corresponding homologs in each species. Warmer color represents higher number of homologs. The bar chart above represents the total number of genes in the corresponding species. **b** Expression patterns of type II MADS-box genes from various organs (from left to right: leaf, androecial lobes, anther and pistil) of *C. sessilifolius*. Expression values were scaled by $\log_2(\text{FPKM} + 1)$, in which FPKM is fragments per kilobase of exon per million mapped reads. **c** Terpenoid biosynthesis (MVA and MEP pathways) related genes in *C. sessilifolius*, and the expression level of each gene was transformed to Z-score across different tissues. Warmer colors indicate high expression levels in both **b** and **c**. Source data are provided as a Source Data file.

specifically enriched (Supplementary Fig. 29 and Supplementary Table 15). A total of 112 genes related to the terpenoid synthesis[60,61] (mevalonate [MVA] pathway and methylerythritol 4-phosphate [MEP] pathway) were further identified. We found that *DXS* and *HMGR* gene families showed more copies than the other five species (*Amborella trichopoda*, *Arabidopsis thaliana*, *Litsea cubeba*, *Nymphaea colorata*, and *Oryza sativa*) (Supplementary Tables 16 and 17). As for the large *TPS* (terpene synthase) gene family, we found that *C. sessilifolius* contained the most members of the *TPS-c* subfamily (Supplementary Fig. 30) responsible for diterpene synthesis[62]. The origin of these expanded genes in *C. sessilifolius* was further examined. We found that their expansions were derived mainly from WGD and tandem duplication (Supplementary Figs. 31 and 32). We also found that most genes showed high expressions in both flower organs and phloem. The genes involved in the final three steps of the sesquiterpenes biosynthesis only showed high expressions in flower organs (Fig. 5c). The *IPPI* (Isopentenyl pyrophosphate isomerase) and *FPPS* (farnesyl diphosphate synthase) genes showed the highest expression in the lobe. These two genes could bidirectionally catalyze the conversion between isopentenyl diphosphate and dimethylallyl diphosphate[63] and synthesis sesquiterpene precursors[64]. The *TPS-a* subfamily members are mainly responsible for sesquiterpene synthesize in the final step[65], and most genes showed high expressions in the pistil (Supplementary Fig. 33).

*C. sessilifolius* also had a special secondary cell wall (SCW) formation with only scalariform perforation plates, which is the primary form of vessel elements. We then focused on the analysis of NAC domain transcription factors, which are critical in SCW biosynthesis with diverse roles in plant development and stress responses[66–68]. A total of 109 *NAC* genes were identified in *C. sessilifolius*, more than those in *Amborella trichopoda* (45) and *Arabidopsis thaliana* (81), but fewer than those in *Oryza sativa* (141) and *Litsea cubeba* (112) (Supplementary Fig. 34). Vascular-related NAC-domains (VNDs)[69–71] and NAC Secondary Wall Thickening Promoting Factors (NSTs)[72–74] are crucial for secondary cell wall biosynthesis. We found only one orthologous of *VND7* for *C. sessilifolius*, similar to *A. trichopoda*, while four *NSTs* copies for *C. sessilifolius* (Supplementary Fig. 34). We also identified one copy of *secondary wall-associated VND-Interacting protein* (*VNI*) and two copies of *secondary wall-associated NAC-domains* (*SNDs*) in the *C. sessilifolius* genome. These genes are critical for regulating SCW biosynthesis[70,71]. Most of these genes showed wide expressions in different tissues, while *VND7*, *NST4*, *SND2*, and *SND3* showed the highest expressions in the xylem (Supplementary Fig. 35). All SCW genes are found in *C. sessilifolius*. The formation of primary vessel elements in this species may follow a more complex process that depends on fine regulations of these genes and others[75,76].

In summary, we provide a high-quality chromosome-level *C. sessilifolius* genome assembly by combining Nanopore, Illumina, and Hi-C sequencing. This fills the genomic gap for one of the major angiosperms lineages Chloranthales and provides a valuable genomic foundation for gaining a deeper understanding of early angiosperm diversification. One independent whole genome duplication was detected within *C. sessilifolius* and the polyploidization events in each Mesangiospermae lineage were mutually independent. Our phylogenetic analyses suggested that Chloranthales and magnoliids are sister groups and they are together sister to eudicots + Ceratophyllales. We found that both hybridization and ILS may have contributed to the strong discordance among gene trees between these lineages. Further comparisons of MADS-box genes suggest that most (especially ABC ones) show non-tissue-specific ancestral functions. The expanded gene families mainly involved in the terpenoid

biosynthesis may partly account for the rich volatile organic compounds in *C. sessilifolius*. All SWC genes are found in this species and its primitive vessel element may have developed through finer genetic regulation rather than gene loss. In summary, the genome sequence for Chloranthales will strongly facilitate future comparative investigations of genic and genomic evolution that underpin the morphological, physiological, and ecological diversification of angiosperms.

## Methods

**Sample collection and sequencing.** Fresh leaf tissues were sampled from a wild *C. sessilifolius* plant growing in Mount Emei, Sichuan Province, China, and immediately stored in liquid nitrogen (Supplementary Fig. 1). All samples were sent to Grandomics (Wuhan, China) for genomic sequencing. The high molecular weight genomic DNA was firstly prepared by the CTAB method and purified with QIAGEN® Genomic kit (Cat#13343, QIAGEN). For the Illumina short reads, the DNA libraries with 500 bp insert sizes were constructed and sequenced using an Illumina HiSeq 4000 platform. For the long-read, the genomic libraries with 20 Kb insertions were constructed and sequenced utilizing a PromethION instrument (Oxford Nanopore Technologies). The raw reads were filtered using the common criteria (presence of adapter, low-quality bases and "mean_qscore <7"). The Hi-C (high-throughput chromosome conformation capture) sequencing was performed as follows: sampled DNA was cross-linked with 1% formaldehyde to capture the interacting DNA segments, chromatin was digested with the Dpn II restriction enzyme, then libraries were constructed and sequenced using the Illumina HiSeq 4000 platform.

For transcriptome analysis, leaf, xylem, phloem, lobe, anther, and pistil of *C. sessilifolius* were collected with three replicates for each tissue on 16th April 2021. Total RNA extraction, library construction and sequencing were performed by BGI-Shenzhen Company (Wuhan, China) on the MGI2000 platform by 2 × 150 bp pair-end model (Supplementary Table 1).

**Genome size estimate and assembly.** To estimate the genome size of *C. sessilifolius*, we surveyed 150 bp paired-end reads, computed 21 bp K-mer frequencies using Jellyfish[77], and exported the resulting histogram into findGSE[78]. Nextdenovo (https://github.com/Nextomics/Nextdenovo) was selected for correcting reads with parameters "read_cutoff=2k, seed_cutoff=30k, blocksize=1.5g" and then Smartdenovo (https://github.com/ruanjue/smartdenovo) for de novo assembly with parameters "wtpre -J 3,000; wtzmo-k 21 -z 10 -Z 16 -U -1 -m 0.1 -A 1000; wtclp -d 3 -k 300 -m 0.1 -FT; wtlay -w 300 -s 200 -m 0.1 -r 0.95 -c 1". The preliminary contigs were further polished by aligning the Illumina short reads to the contigs using Nextpolish[79]. After four rounds of successive iterative correction, the final genome sequence was obtained. The GC content and sequencing coverage analyses were applied to evaluate the presence of contamination. The quality of the genome assembly was also assessed using BUSCO[80] (Benchmarking Universal Single-Copy Orthologs) with the embryophyta_odb10 database. The clean Hi-C data were mapped to contig sequences by Bowtie2[81] and 354 Mb valid interaction pairs were extracted. Based on those chromatin interactions, LACHESIS[82] was employed to cluster, order, and orient the contigs into pseudo-chromosomes.

**Repeat and gene prediction.** RepeatModeler (http://repeatmasker.org/RepeatModeler.html) was applied initially to build a de novo repeat library. The library and a known repetitive elements database (Repbase, http://www.girinst.org/repbase) were used to detect repetitive sequences by RepeatMasker[83] (http://repeatmasker.org/) with default parameters. In addition, we ran LTR_retriever[84], which integrated results of LTRharvest[85] and LTR_FINDER[86], to identify the LTR-RTs from the whole genome. The insertion time of LTRs was also calculated by LTR_retriever using the Eq. 1

$$T = K/2r, \tag{1}$$

where $K$ is the genetic distance and $r$ is the mutation rate of repeat sequences. We inferred the synonymous substitution in coding regions of SSCGs dataset ($1.9 \times 10^{-9}$ per site per year) using r8s[87]. We used 2-fold higher rate[88] ($3.8 \times 10^{-9}$ per site per year) to represent the mutation rate of repeat sequences. To infer the protein-coding genes of the *C. sessilifolius* genome, an annotation strategy that combined homology-based prediction, ab initio prediction and transcriptomic evidence was applied. Homologous gene sets from seven reference genomes (*A. trichopoda*, *Ar. thaliana*, *Ci. kanehirae*, *O. sativa*, *Pr. persica*, *V. vinifera*, and *Zea mays*) were searched against the genome by GeMoMa[89], then three programs (Augustus[90], Genscan[91], and GlimmerHMM[92]) were used for de novo prediction. The de novo assembled transcripts by Trinity[93] were also aligned to the genome to generate the transcriptome evidence by PASA[94]. The results generated from these approaches were integrated into the final consensus gene sets using the EvidenceModeler pipeline[95]. For functional annotation, InterProScan[96], NCBI non-redundant protein database (NR) and SwissProt database were used and searched by BLASTP[97].

**Polyploidization analysis**. The toolkit WGDI[98] was selected to infer the polyploidization history of 11 species. Collinear genes were firstly identified with the parameter "-icl" of WGDI within each genome and between genomes, and the collinear genes dot plots were used to count the syntenic ratios between different species to confirm the polyploidy level of each species. Frequencies of synonymous substitutions per synonymous site (Ks values) between colinear genes were estimated using the Nei-Gojobori approach as implemented in PAML[99]. The median Ks values of each block were selected to perform the Ks peak fitting by WGDI with the parameter "-pf".

We further applied the collinear gene phylogenomic analysis to check if the WGD occurred independently within the selected six species. The collinear genes were extracted by WGDI (-at) and used to infer maximum likelihood (ML) trees by IQ-TREE[100] with automatic selection of the best-fit substitution model (-m MFP) and 1000 ultrafast bootstrap replicates (-bb 1000). ASTRAL[101] could calculate the frequencies of collinear genes trees that support independent WGD event in each species-pair. For example, within the two species A and B (assuming each has one WGD), the extracted collinear genes are named as A1, A2, B1, and B2. So the topology of ((A1, B1), (A2, B2)) supporting WGD event is shared by A and B, while only the topology of ((A1, A2), (B1, B2)) supporting WGD event occurring in A and B are independent. So, we used ASTRAL with the parameter "-t 2" and the specific tree ((A1, A2), (B1, B2)) to calculate the number of genes that support independent WGD. If the WGT occurred within A and B, the topology may appear as "(((A1, A2), A3)#, ((B1, B2), B3)));", and the supporting frequency of the internal branch can be marked as "#", which represents the occurrence of independent WGT in A and B.

**Phylogenetic analysis**. Three sets of homologous genes (SSCGs, OSCGs, and LCGs) were generated by analyzing genomes of the 14 species representing major lineages of angiosperms to infer the phylogenetic placement of *C. sessilifolius* (Supplementary Table 11). SSCGs represent the single-copy genes identified using SonicParanoid[42] with default parameters among 14 species (*Aquilegia coerulea, Apostasia shenzhenica, Amborella trichopoda, Ceratophyllum demersum, Cinnamomum kanehirae, Chloranthus sessilifolius, Euryale ferox, Elaeis guineensis, Ginkgo biloba, Liriodendron chinense, Nymphaea colorata, Oryza sativa, Prunus persica,* and *Vitis vinifera*). OSCGs represent the single-copy genes identified with OrthoMCL[102] with default parameters among 14 species mentioned above and a Gymnosperm species (*Picea abies*), and each cluster allows up to two species to be missing. LCGs represent the low-copy genes, which ranged between one and five gene copies per cluster, and were identified among 14 species by OrthoMCL. Concatenation and coalescent approaches were applied to reconstruct phylogenetic trees. Each of the acquired amino acid sequences was first aligned and trimmed using MAFFT[103] and Phyx[104], respectively. The resulting sequences were converted into corresponding codon alignments by PAL2NAL[105]. Subsequently, for the concatenation approach, sequences of genes in each dataset (except LCGs) were concatenated using an in-house Python script. The concatenation tree and each cluster gene tree were constructed by IQ-TREE[100] (-m MFP –bb 1000) and the coalescent tree was inferred by ASTRAL[101]. We further employed STAG[106] and ASTRAL-pro[107] to infer species trees based on the low-copy genes set (LCGs). Besides, TreeShrink[44] was further selected to reduce the influence of long branch attraction in the SSCG and OSCG datasets, which could identify and remove sequences that lead to unrealistically long branch lengths within each cluster. Then, the retained sequences were used to construct the concatenation and coalescent trees with the same method mentioned above. To further eliminate errors in orthology inference, we used the synteny relationship to identify the orthologous genes by WGDI, which don't need gene family clustering. A total of 11 species mentioned above were selected and we excluded *Ginkgo biloba, Apostasia shenzhenica,* and *Oryza sativa* because they don't have chromosome-level assembly or contain complicated polyploidization history. We identified the intergenomic synteny blocks between the reference species *Amborella* and others, and the intragenomic synteny blocks among each species. According to the similarity (estimated by Ks and blast score) and completeness (covered genes and gene span length) of each block, WGDI (-bi and -a) could assign different synteny blocks into different putative sets and mark them in different colors (Supplementary Figs. 15–17). For example, eight synteny blocks were identified in *Ceratophyllum demersum* for each *Amborella* segment, and WGDI assigned each block into eight sets with following colors: red, pink, green, light green, blue, light blue, yellow, and black. Each color represented one set and was respectively named as *Ceratophyllum demersum* 1 to *Ceratophyllum demersum* 8. Each set was considered as one species and used for the phylogenetic analyses. Finally, ASTRAL was used to infer the topology among the different sets of all species. A total of 4120 collinear genes that have a collinear relationship with *Amborella* and have at least eight species were retrieved to infer the collinear gene tree by IQ-TREE, and finally, the synteny-based species tree was constructed by ASTRAL. In addition, to eliminate potential errors during parse taxon sampling, we performed the expanded taxon sampling analysis. A total of 41 species that covered 30 angiosperm orders and one Gymnosperm species (*Ginkgo biloba*) were selected and BLASTP[97] and OrthoMCL[102] were used to group the sequence into different clusters. Each gene cluster was required to include sequences from more than 80% species, and the "mostly" single-copy orthologous genes were identified using a tree-based method[108]. For each gene

cluster, the sequence was aligned by MAFFT[103] and PAL2NAL[105] as described above, and species trees were inferred by ASTRAL.

Divergence times were estimated based on SSCGs using MCMCTree in the PAML package, calibrated with four fossil constraints selected from the TimeTree website (http://www.timetree.org): 330–289 Mya between *G. biloba* and *A. trichopoda*, 199–173 Mya between *A. trichopoda* and *N. colorata*, 163–145 Mya between *Apostasia shenzhenica* and *Ceratophyllum demersum*, and 135–107 Mya between *Prunus persica* and *Vitis vinifera*.

We also reconstructed plastid trees, as follows. GetOrganelle[109] was selected to de novo assemble the complete chloroplast genome of *C. sessilifolius* with the Illumina sequencing reads, and then the genome was annotated with the online program GeSeq[110]. Chloroplast genes of *C. sessilifolius* and published sequences for the 13 other species were aligned as described above, and then concatenated to construct the ML tree by IQ-TREE[100] with "-bb 1000 -MFP".

For visualizations of gene tree discordance, quartet scores were first calculated to evaluate three alternative topologies using ASTRAL. Then DensiTree[111] superimposed all gene trees for the SSCGs, which strongly colored areas with topological uncertainty. We also combined seven taxa into 21 "splits" to depict the portion of gene trees that supported or rejected each hypothesis using DiscoVista[112].

**Hybridization inference and ILS simulation**. Hybridization was detected for the dataset SSCG using the maximum pseudolikelihood estimation of phylogenetic networks, as implemented in PhyloNetworks[47]. Seven species were selected to represent the major lineages from all the 14 species to reduce the software running time, and the selected species were *A. trichopoda* (Amborellales), *N. colorata* (Nymphaeales), *O. sativa* (monocots), *L. chinense* (magnoliids), *C. sessilifolius* (Chloranthales), *Ceratophyllum demersum* (Ceratophyllales), and *V. vinifera* (eudicots). The maximum number of hybridizations was allowed as three times (ranging from hmax = 0 to hmax = 3), and each with 100 runs to ensure accuracy. For the ILS analyses, we first calculated the theta parameter by mutation units inferred by IQ-TREE/coalescent units inferred by ASTRAL, which could reflect the level of ILS (high theta value means large ancestor population size and hence high ILS level)[48]. The Phybase[49] and DendroPy were selected to simulate gene trees under the ILS condition, which is widely used to explain the incongruence within gene trees[17,48,113,114]. They both use the estimated species tree with branch lengths measured in coalescent units as an input, and then simulate the gene trees under the multispecies coalescent model by considering the existence of ILS. The internal branch lengths of the ASTRAL tree were used for simulation, and all terminal branches were set to 1 (as 1 allele was generated for each species). In addition, as theta ranged from 0.027 to 0.224, we also performed another simulation with theta as 0.001 (two hundred times less than the minimal observed theta value) in Phybase to represent the absence of ILS (or extremely low ILS). A total of 20,000 gene trees were generated for each simulation, and then we performed the gene-tree quartet frequencies analyses for each four-species group among all the 14 species. For examples, a four-species group "A, B, C, D" contains three possible topologies: "((A, B), (C, D))", "((A, C), (B, D))" and "((A, D)(B, C))". Then we calculated all gene frequency of all the four-species groups within the simulated and observed gene tree datasets, and used the linear regression model ("lm()") in R to calculate correlations between them.

**Flower development genes analysis**. MADS-box genes were identified using the HMMER[115] and iTAK[116] software, and the parameters "–cut_tc" and Pfam profiles (PF00319) were used for HMM searching. For the other FDRGs, we performed the protein sequences similarity search by BLASTP with an *E*-value cut-off of $10^{-5}$ using the known flowering genes in *Arabidopsis* as a reference. InterProScan[96] was applied to further check the integrity of candidate gene domains. Multiple sequence alignment and ML tree inference were performed to group them into subfamily, and the genes set that clustered with reference sequence was used for the next analysis. For the transcriptome analysis, the clean RNA-seq reads of the six tissues were mapped onto *Chloranthus sessilifolius* genome using HISAT2[117]. StringTie[118] was then used to calculate fragments per kilobase of transcript per million mapped reads (FPKM values) for each sample. The reproducibility among the biological replicates was further evaluated by the multidimensional scaling plot and the Pearson correlation analysis (Supplementary Fig. 27).

**Terpenoid and secondary cell wall biosynthesis genes analysis**. Genes related to terpenoid backbone biosynthesis (including MVA pathway and MEP pathway) were retrieved from *Arabidopsis thaliana* (https://www.arabidopsis.org/). These proteins were then used to search for homologs in the predicted proteome of *C. sessilifolius* using BLASTP with the *e*-value of 1e-5 and identity value >40%. Conserved domains of the TPS gene family (PF01397 and PF03936) and NAC gene family (PF02365) were used to search against the proteome using hmmsearch. Phylogenetic analysis was performed as described above.

**Reporting summary**. Further information on research design is available in the Nature Research Reporting Summary linked to this article.

## Data availability

All of the raw sequence reads used in this study and the genome assembly have been deposited at NCBI under the BioProject accession number PRJNA759285. We also uploaded the the assembly and annotation files in the Genome Warehouse in BIG Data Center under the BioProject accession number PRJCA006913. The annotation files (gff, CDS, and proteins) are also available at GitHub [https://github.com/yongzhiyang2012/Chloranthus-sessilifolius-genome/tree/main/Annotation]. Source data are provided with this paper.

## Code availability

All the custom scripts and specific command lines have been deposited in GitHub [https://github.com/yongzhiyang2012/Chloranthus-sessilifolius-genome].

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

## Acknowledgements

Financial support was equally provided by the Strategic Priority Research Program of Chinese Academy of Sciences (XDB31000000 to J.L. and Y.Y.), the Second Tibetan Plateau Scientific Expedition and Research (STEP) program (2019QZKK0502 to J.L.), and the National Natural Science Foundation of China (31590821 to J.L.). Further supports were provided by National Key Research and Development Program of China (2017YFC0505203 to J.L.) and International Collaboration 111 Programme (BP0719040). All the computation works were supported by the Big Data Computing Platform for Western Ecological Environment and Regional Development, and the Supercomputing Center of Lanzhou University.

## Author contributions

Y.Y. was the leader of this study and designed the experiments and coordinated the project. J.M., P.S., D.W. and C.D. performed field work and collected samples. J.M. and D.W. performed the genome assembly. J.M., D.W., J.Y., Y.L. and W.M. carried out the genome annotation and phylogenomic analyses. P.S., Z.W., R.X. and Y.W. performed the polyploidization analyses. J.M. and D.W. performed the transcriptome analyses. Y.Y., J.M. and J.L. wrote the manuscript and N.S. polished the English writing. All of the authors read and approved the final manuscript.

## Competing interests

The authors declare no competing interests.
