## [Peer Review File · Nature Communications]

The *Chloranthus sessilifolius* genome provides insight into early diversification of angiospermsEditorial Note: This manuscript has been previously reviewed at another journal that is not operating a transparent peer review scheme. This document only contains reviewer comments and rebuttal letters for versions considered at Nature Communications.

Reviewers' Comments:

Reviewer #1:

Remarks to the Author:

The authors have substantially revised the manuscript and carried out additional analyses to bolster their conclusions. In all, I believe this manuscript is now much more thorough and most of my previous questions/concerns are addressed. That said, I still have a few comments and suggestions.

(1) Tissue specificity. The authors stated that "the two AP3s are weakly expressed in all the examined tissues, and PI is broadly expressed in both flowers and leaves". However, from Fig. 5b, it is clear to me that AP3_1 is expressed at a much higher level in anthers than in other tissues (especially leaves, xylem and phloem). The same is true for PI but less striking. Also note the log₂ scale. So I'm not sure why the authors concluded that these genes lack tissue specificity. It is also worth mentioning that many of the ABC genes can be found in eudicot 1KP transcriptomes even though RNA came from leaf tissues.

(2) Aristolochia. An Aristolochia genome was very recently published:

<https://www.nature.com/articles/s41477-021-00990-2>. While I don't believe it is fair to ask the authors to redo all their analyses in light of this new genome, I do think the Aristolochia paper, which presents new data on phylogeny and WGD, should be discussed.

Line 46. The use of "Similarly" is a bit odd, given the topology is different from what was described in the previous sentence.

Line 48. Change "genomes" to "genomic".

Line 49-52. This sentence is not connected very well with the previous sentences. Please smooth it out.

Line 58. Should it be "three stamens and one pistil"?

Line 61. Italicize the genus names.

Line 64. Change "unclear evolution and relationship" to something like "resolve the evolutionary relationship".

Line 70-72. "Our analyses..." this sentence is not well-constructed. Something is missing before "identify".

Line 86. Change "GC and depth" to something like "GC content and sequencing coverage".

Line 93. ":" to ",".

Line 113. How was the divergence time calculated? In the method it was mentioned that LTR_retriever was used but as far as I know LTR_retriever does not have this function.

Line 119. Change "blocks" to "syntenic blocks".

Line 276. Change "expansion genes" to "expanded gene families"

Line 287. Change "origin patterns" to "origin".

Line 324-325. I'm not sure I agree with this, given Fig. 5b. See my comment above.

Line 329. Remove "acquired".

Line 360. Should it be "GC content and sequencing coverage"?

Line 383. "user-friendly" is not necessary here.

Line 399. Is this ASTRAL-PRO?

Line 432. Is this ASTRAL-PRO? Essentially you have multi-labeled gene trees, which are not compatible with ASTRAL. Please elaborate how exactly you ran the ASTRAL analysis here.

Figure 2b. heatmap legend, "independently Polyploidization" to "independent polyploidization".

Figure 4. I'm confused about c-f. Are you sure the y-axis is labeled correctly in (e) and (f)? Shouldn't it be "DendroPy with ILS"?

Supplementary Figure 7. Please spell out genus names in (a).

REVIEWER COMMENTS

Reviewer #1 (Remarks to the Author):

The authors have substantially revised the manuscript and carried out additional analyses to bolster their conclusions. In all, I believe this manuscript is now much more thorough and most of my previous questions/concerns are addressed. That said, I still have a few comments and suggestions.

(1) Tissue specificity. The authors stated that “the two AP3s are weakly expressed in all the examined tissues, and PI is broadly expressed in both flowers and leaves”. However, from Fig. 5b, it is clear to me that AP3_1 is expressed at a much higher level in anthers than in other tissues (especially leaves, xylem and phloem). The same is true for PI but less striking. Also note the log₂ scale. So I’m not sure why the authors concluded that these genes lack tissue specificity. It is also worth mentioning that many of the ABC genes can be found in eudicot 1KP transcriptomes even though RNA came from leaf tissues.

Reply: Thank you for pointing this out. Based on your comment, we have corrected the description of AP3 genes. Because calyx and petal are absent in *Chloranthus sessilifolius*, we couldn’t identify more expression limitations of ABC genes in different flower organs. So, we have rewritten this section to reduce emphasis on the lack of tissue specificity of ABC genes. Also, following your suggestion, we added a brief discussion about 1KP transcriptomes. (Lines 273-281)

(2) Aristolochia. An Aristolochia genome was very recently published: <https://www.nature.com/articles/s41477-021-00990-2>. While I don’t believe it is fair to ask the authors to redo all their analyses in light of this new genome, I do think the Aristolochia paper, which presents new data on phylogeny and WGD, should be discussed.

Reply: Thanks for suggesting this paper. This paper was also into our notice. In fact, the study performed many phylogenomics analyses that shed light on the evolution of angiosperms. Especially, they identified fusions that supported a sister relationship between magnoliids and monocots, which is immensely valuable for future researches. In the revision, we have cited this paper and added more discussion in the introduction as well as results & discussion. (Lines 50-52, 147, 164-168)

Line 46. The use of “Similarly” is a bit odd, given the topology is different from what was described in the previous sentence.

Reply: We have changed the sentence to “The OneKP Project was also based on transcriptome data but identified sister relationship between Chloranthales and magnoliids.” (Lines 46-48)

Line 48. Change “genomes” to “genomic.

Reply: Done. (Line 48)

Line 49-52. This sentence is not connected very well with the previous sentences. Please smooth it out.

Reply: Thanks for your suggestion. We have rephrased this sentence to connect it well with the earlier sentences. (Lines 52-55)

Line 58. Should it be “three stamens and one pistil”?

Reply: Thanks, we have rewritten it. (Line 61)

Line 61. Italicize the genus names.

Reply: Done. (Line 64)

Line 64. Change “unclear evolution and relationship” to something like “resolve the

evolutionary relationship”.

Reply: Done. (Line 67)

Line 70-72. “Our analyses...” this sentence is not well-constructed. Something is missing before “identify”.

Reply: Thanks for your suggestion. We have rephrased this sentence to make it clearer. (Lines 73-75)

Line 86. Change “GC and depth” to something like “GC content and sequencing coverage”.

Reply: Done. (Line 89)

Line 93. “:” to “;”.

Reply: Done. (Line 96)

Line 113. How was the divergence time calculated? In the method it was mentioned that LTR_retriever was used but as far as I know LTR_retriever does not have this function.

Reply: We ran the whole LTR_retriever analyses based on the author recommendation (https://github.com/oushujun/LTR_retriever), and the Module 4 was developed for estimating the insertion time of each intact LTR-RTs (Ou et al. 2018). It was based on the divergence of solo-LRTs in each complete LTRs, and scaled to insertion time based on the mutation rate. The results were listed in the output file of “Cse_genome.fa.pass.list”, and all the parameters were set as default (command line: LTR_retriever -genome Cse_genome.fa -inharvest Cse_genome.fa.finder.combine.scn -threads 60).

Ou, S., & Jiang, N. (2018). LTR_retriever: a highly accurate and sensitive program for identification of long terminal repeat retrotransposons. *Plant physiology*, 176(2), 1410-1422.

Line 119. Change “blocks” to “syntenic blocks”.

Reply: Done. (Line 121)

Line 276. Change “expansion genes” to “expanded gene families”

Reply: Done. (Line 287)

Line 287. Change “origin patterns” to “origin”.

Reply: Done. (Line 298)

Line 324-325. I’m not sure I agree with this, given Fig. 5b. See my comment above.

Reply: We have revised this sentence to make it clearer. (Lines 273-281)

Line 329. Remove “acquired”.

Reply: Done.

Line 360. Should it be “GC content and sequencing coverage”?

Reply: Yes, we have changed them based on your suggestion. (Line 368)

Line 383. “user-friendly” is not necessary here.

Reply: We have deleted this word.

Line 399. Is this ASTRAL-PRO?

Reply: We did not use ASTRAL-PRO but used ASTRAL instead. ASTRAL-PRO is used for non-single copy gene families, and for generating the final species tree with one species one tip. In our study, for each collinear gene pair from the two species A and B that both have one WGD, we randomly marked them as “A1, A2” and “B1, B2”. The gene trees contain three topologies: T1 “((A1,A2),(B1,B2));”, T2 “((A1,B1),(A2,B2));”, T3 “((A1,B2),(A2,B1));”. So, we just calculated the frequency of T1 by ASTRAL to represent the possibility of

independent duplication in A and B. If the WGT occurred within A and B, we specify the topology as “(((A1,A2),A3)#,((B1,B2),B3)))”, and only focus on the supporting frequency of the internal branch marked as “#”, which represent the occurrence of independent WGT in A and B. We have added more description in revised manuscript to make it clearer. (Lines 408-410)

Line 432. Is this ASTRAL-PRO? Essentially you have multi-labeled gene trees, which are not compatible with ASTRAL. Please elaborate how exactly you ran the ASTRAL analysis here.

Reply: It is not again the ASTRAL-PRO but the ASTRAL, and we have added more details in the revised manuscript to make it clearer. Specifically, we added following details in the methods: “We identified the intergenomic synteny blocks between the reference species *Amborella* and others, and the intragenomic synteny blocks among each species. According to the similarity (estimated by Ks and blast score) and completeness (covered genes and gene span length) of each block, WGDI (-bi and -a) could assign different synteny blocks into different putative sets and mark them in different colors (shown in Supplementary Figs. 15-17). For example, eight synteny blocks were identified in *Ceratophyllum demersum* for each *Amborella* segment, and WGDI assigned each block into eight sets with following colors: red, pink, green, light green, blue, light blue, yellow and black. Each color represented one set and was respectively named as *Ceratophyllum demersum* 1 to *Ceratophyllum demersum* 8. Each set was considered as one species and used for the phylogeny analyses. Finally, ASTRAL was used to infer the topology among the different sets of all species.” We have now added more description in method to remove any ambiguity, and we also slightly modified the Supplementary Fig. 18 to more clearly show our process. (Lines 438-448)

Figure 2b. heatmap legend, “independently Polyploidization” to “independent

polyploidization”.

Reply: Changed as suggested.

Figure 4. I’m confused about c-f. Are you sure the y-axis is labeled correctly in (e) and (f)? Shouldn’t it be “DendroPy with ILS”?

Reply: We are sorry about the misspelled labels. The simulation performed by DenderoPy was with ILS and we have corrected them in the revised manuscript.

Supplementary Figure 7. Please spell out genus names in (a).

Reply: Done.

Reviewers' Comments:

Reviewer #3:

Remarks to the Author:

The authors have addressed my previous questions nicely. Still two more comments (Sorry!).

(1) To calculate LTR insertion time, you'd need a mutation rate right? How was this rate estimated? Simply taking a rate from, say maize, and apply here is not appropriate. In my opinion, calculating LTR insertion time needs more rigorous modelings than what are typically done these days.

(2) line 284. "more expression speciality of the ABC genes remains unknown." I'm unsure what this means.

Reviewer #3 (Remarks to the Author):

The authors have addressed my previous questions nicely. Still two more comments (Sorry!).

(1) To calculate LTR insertion time, you'd need a mutation rate right? How was this rate estimated? Simply taking a rate from, say maize, and apply here is not appropriate. In my opinion, calculating LTR insertion time needs more rigorous modelings than what are typically done these days.

Reply: We used the mutation rate during the “LTR insertion time” analyses and we agree that the mutation rate will influence the insertion time as it was calculated using the formula $T = K/2r$ (K is the genetic distance and r is the mutation rate of repeat sequences). Directly calculating the mutation rate of repeat sequences is difficult. Therefore, many papers have unanimously used the reported repeat mutation rate of rice (Ma & Bennetzen, 2004) to calculate the insertion times of LTRs. Ma & Bennetzen (2004) firstly calculated the mutation rate of coding regions and then used 2-fold higher rate to represent the repeat mutation rate. It has been proved that the repeat mutation rate is usually 2-fold higher than the coding regions mutation rate. Therefore, we also followed this approach to infer the mutation rate. We firstly estimated the mutation rate of coding region by r8s and SSCG dataset (1.9×10^{-9} per site per year), and then scaled it to the mutation rate of repeat sequences (3.8×10^{-9} per site per year). Based on this mutation rate and the genetic distance of repeat sequences calculated by LTR_retriever, we updated the LTR insertion times and added the detail description in the revised manuscript (Lines: 113-116 [only changed the insertion time], 380-384 and Supplementary Fig. 7d).

Ma, J., & Bennetzen, J. L. (2004). Rapid recent growth and divergence of rice nuclear genomes. *Proceedings of the National Academy of Sciences*, 101(34), 12404-12410.

(2) line 284. "more expression speciality of the ABC genes remains unknown." I'm unsure what this means.

Reply: We apologize for this ambiguous sentence. We just wanted to point out that we couldn't identify more expression patterns of ABC genes in the classic flower organs, as C.

sessilifolius lacks sepal and petal. While, this sentence is not directly related to our manuscript, we deleted it in the revised manuscript.

Reviewers' Comments:

Reviewer #3:

Remarks to the Author:

I don't have any other comment. Kudos to the authors for putting this paper together.

Reviewer #3 (Remarks to the Author):

I don't have any other comment. Kudos to the authors for putting this paper together.

Reply: Thank you for your efforts to help us improve the quality of our manuscript.